# Electrospun Nanofiber Covered Polystyrene Micro-Nano Hybrid Structures for Triboelectric Nanogenerator and Supercapacitor

**DOI:** 10.3390/mi13030380

**Published:** 2022-02-26

**Authors:** Jihyeon Park, Seungju Jo, Youngsu Kim, Shakir Zaman, Daewon Kim

**Affiliations:** 1Department of Electronics and Information Convergence Engineering, Kyung Hee University, 1732 Deogyeong-daero, Giheung-gu, Yongin 17104, Korea; jihyeon.park@khu.ac.kr (J.P.); joseungju@khu.ac.kr (S.J.); youngsukim@khu.ac.kr (Y.K.); shakirzaman@khu.ac.kr (S.Z.); 2Institute for Wearable Convergence Electronics, Kyung Hee University, 1732 Deogyeong-daero, Giheung-gu, Yongin 17104, Korea; 3Department of Electronic Engineering, Kyung Hee University, 1732 Deogyeong-daero, Giheung-gu, Yongin 17104, Korea

**Keywords:** polystyrene, electrospinning, micro-nano structure, triboelectric nanogenerator, energy storage

## Abstract

Recently, tremendous research on small energy supply devices is gaining popularity with the immerging Internet of Things (IoT) technologies. Especially, energy conversion and storage devices can provide opportunities for small electronics. In this research, a micro-nano structured design of electrodes is newly developed for high performing hybrid energy systems with the improved effective surface area. Further, it could be simply fabricated through two-steps synthesis of electrospinning and glass transition of a novel polystyrene (PS) substrate. Herein, the electro-spun nanofiber of polyacrylonitrile (PAN) and Nylon 66 (Nylon) are applied to the dielectric layer of a triboelectric generator (TENG), while the PAN and polyaniline (PANI) composites is utilized as an electroactive material of supercapacitor (SC). As a result, the self-charging power system is successfully integrated with the wrinkled PAN/PS (W-PAN/PS@PANI)-SC and W-TENG by using a rectifier. According to the fabricated hybrid energy systems, the electrical energy produced by W-TENG can be successfully stored into as-fabricated W-PAN/PS@PANI-SC and can also turn on a commercial green LED with the stored energy. Therefore, the micro-nano structured electrode designed for hybrid energy systems can contribute to improve the energy conversion and storage performance of various electronic devices.

## 1. Introduction

Recently, the severe problems in energy supply have been widely drawing attention due to the use of small devices and internet of things (IoT) devices. As the energy crisis has become a global problem, numerous efforts have been dedicated to treat these issues by producing sustainable and renewable green energy [1], such as thermal [2], solar [3], and wind [4]. Additionally, in the development of small electronic devices, it is essential to treat the limitation of device miniaturization due to the existing energy systems with complicated external circuits and power management. However, an energy harvesting system based on mechanical energy conversion can sufficiently satisfy the need of small portable electronics [5]. Among these energy harvesting techniques, the triboelectric nanogenerator (TENG) has the numerous advantages of simple structure, various materials, low cost, and applicability to various energy sources [6,7,8,9]. Additionally, the TENG can generate electrical energy through a simple contact–separation of two dielectrics materials from external mechanical movements [10,11,12]. In detail, there are four types of operating structures: a two-electrode mode, a single electrode mode, a sliding mode, and free-standing mode [13,14,15,16].

The development of an energy storage system through the supercapacitor (SC) is crucial for the purpose of preventing the waste of generated electrical energy from TENG. Through the utilization of a hybrid energy system combining TENG and SC, the electrical energy produced by TENG can be efficiently stored in the SC, which contains a high-power density and a high charge/discharge cycle [17]. To develop the merged and independent one-module device, numerous researches related to hybrid energy systems have been studied for effective energy efficiency in self-powered energy storage systems [18,19,20].

However, the improvement of output power performance has remained as a main drawback for producing a practical TENG [21,22,23,24]. To achieve the enhanced electrical output of TENG, the most important factor that determines the electrical output performance of TENG is the surface charge density of the dielectric layer. As the surface charge density increases, a larger potential difference is induced. As a method of increasing the surface charge density of a dielectric layer, it is possible to increase the effective contact area of the dielectric layer [25]. Similarly, a significant factor in increasing the capacitance of the SC is the increase of electroactive surface area of the electrode. As the electroactive area increases, the electric double layer capacitor (EDLC) can absorb and desorb a larger number of ions from a higher surface area. In the case of pseudo-capacitors with the electrode materials of metal oxide or conductive polymer, a Faradaic electron-transfer process is induced to store the electrical energy [26,27].

Herein, a wrinkled polystyrene (PS) and electrospun nanofiber-based energy harvesting and storage hybrid system is newly developed. According to the proposed synthetic process, the effective surface area of electrodes could be dramatically increased without using the complex fabrication process. The PS material could create a micro-wrinkled shape through shrinkage from the glass transition by heat [28]. Firstly, Cu (copper) conductive layer was thinly deposited for the conductive PS substrate as an electrode. In this case, when the nanofiber structure created by electrospun-polymer membrane of polyacrylonitrile (PAN) and Nylon 6/6 (Nylon) covers the wrinkled PS material, the nanofiber membrane maintains the nanostructure by forming a micro-nano hybrid structure. As a result, compared to non-wrinkled PS-TENG, the wrinkled PS based TENG (W-PS TENG) showed 13 times higher open circuit voltage (*V*_OC_) and 37.5 times higher short circuit current (*I*_SC_) under an operating frequency of 1 Hz and a compressive force of 10 N. Additionally, the 6.2 μW/cm^2^ of averaged electrical power density was measured at 20 MΩ of optimal load resistance. In addition, to show that the areal capacitance is increased by using the micro-nano hybrid structure to SC substrate, the areal capacitance of SC using PAN nanofiber substrate and W-PS substrate coated with Polyaniline (PANI) as an active material were compared. According to the material properties of PAN, a high porosity, small pore size, large specific surface area, and a light-weight membrane can be fabricated. When manufacturing electrodes use PANI, they have high conductivity, and can be easily synthesized through a solution, allowing coating with little deformation of the surface structure of the substrate. Therefore, PANI-coated substrate can be used as an electrode of SC while maintaining the previously manufactured micro-nano hybrid structure. The wrinkled PS film covered with PAN nanofiber (W-PAN/PS) based SC, the areal capacitance was 3 times and 2.25 times higher than that of the using only PAN nanofiber and only W-PS, respectively. In addition, in the case of a hybrid SC made using carbon fiber as a counter electrode, a power density of 6.25 μW/cm^2^ at an energy density of 2.31 mWh/cm^2^. Therefore, through a simple process of PS substrate and electrospinning, it is possible to improve the electrical output of the TENG as well as the capacitance of SC. Finally, the hybrid energy systems composed of TENG, and SC are connected through a rectifier and configured as one device. The electrical energy produced by the TENG can be successfully stored in the as-fabricated SC for self-powered energy storage systems. The energy stored in the SC was successfully transferred to the load through the switch when the user wanted it and can be used to light an LED. The self-powered energy storage system can store 0.9 V for 115 s when operated at a vertical force of 10 N and a frequency of 5 Hz. Through this, it was shown that the self-powered energy storage system manufactured in this study can be used as a power source for small electronic devices.

## 2. Materials and Methods

### 2.1. Materials

The commercial polystyrene (PS) film with a thickness of 250 μm was utilized as received. Industrial polyacrylonitrile (PAN) with a molecular weight of 150,000 N, N-dimethylformamide (DMF), nylon 6/6 (Nylon), aniline, and ammonium persulfate (APS) were purchased from Sigma-Aldrich (St. Louis, MI, USA). Formic acid was obtained from Daejung-Chemicals (Siheung, Korea).

### 2.2. Deposition of Thin Cu Layer onto PS Substrateaterials

The conductive Cu layer was sputtered onto the PS substrate under 100 W for 30 min.

### 2.3. Preparation of Membrane Layer onto PS Layer

The PAN powder was dissolved into DMF solvent, by magnetically stirring overnight at 50 °C. The nylon pellet was dissolved into formic acid solvent, by magnetically stirring overnight at 50 °C. Then, it was filled into a syringe terminated by a stainless-steel needle. The needle held at 10 kV using an electrospinning machine (Electrospinning system, MTDI, Daejeon, Korea) at a distance of 10 cm from the grounded collector drum. Conductive PS film for TENG and bare PS film for the supercapacitor were attached at collector. The polymer solution was fed at a speed 1.5 mL/h and the drum collector was spun 500 rpm for 1 h.

### 2.4. Physical Characterizations

The surface morphologies of the prepared electrodes were analyzed by field emission scanning electron microscope (FE-SEM, Carl Zeiss, MERLIN, Jena, Germany). The crystalline structure of the prepared electrodes was examined by X-ray diffractometer (XRD, Bruker, D8 Advance, Billerica, MA, USA). The sheet resistance was measured by 4-point probe (CMT-100S, AIT Co., Ltd., Gyeonggi-do, Korea).

### 2.5. Electrical Characterization

An electrodynamic shaker (Labworks, Lw139.138-40, Lehi, UT, USA) was used to generate and apply vibration, to establish periodic contact and separation between the dielectric material and counter triboelectric layer. The open-circuit voltage and short-circuit current generated by the W-TENG were measured by an electrometer (Tektronix, Keithley 6514, Beaverton, OR, USA).

### 2.6. Electrochemical Characterization

The electrochemical performance was measured using a three-electrode system with an electrochemical workstation (IVIUM Technologies, Eindhoven, The Netherlands). In this process, a platinum (Pt) wire and a silver/silver chloride (Ag/AgCl) acted as a counter and reference electrode, respectively. The as-prepared electrode on PS substrate as a working electrode was immersed in 1 M potassium hydroxide (KOH) electrolyte. The electrochemical performance was characterized by the measurement of cyclic voltammetry (CV), galvanostatic charge/discharge (GCD), and electrochemical impedance spectroscopy (EIS). Further, the values of areal capacity (*Q*_A_) and areal capacitance (*C*_A_) were calculated by the following Formulas (1) and (2):(1)$$Q_{A}(\frac{\mathrm{Ah}}{{\mathrm{cm}}^{2}})=\frac{I\times \Delta t}{A},$$
(2)$$C_{A}=\frac{I\times \Delta t}{A\times \Delta V}$$
where, *Q*_A_ is the specific capacity (Ah cm^−2^), and *C*_A_ is the specific capacitance (F cm^−2^). *I* is the applied discharge current (A), ∆*t* is the discharge time (s), A is the area of electroactive material (cm^2^), and ∆*V* is the potential window (*V*).

### 2.7. Fabrication HSC

The hybrid supercapacitor (HSC), composed of a positive electrode, negative electrode, separator, and electrolyte, was fully sealed for two electrode configurations. The suggested PANI-coated PAN nanofiber with W-PS substrate was used as a battery-type electrode (+ve electrode) and carbon fabric as a capacitive electrode (−ve electrode) for the assembly of HSC. These two electrodes were separated by the filter paper and immersed in 1 M KOH electrolyte. Consequently, the *C*_A_, areal energy density (*E*_AD_), and power density (*P*_D_) of the fabricated HSC were estimated using the following Formulas (3) and (4):(3)$$E_{\mathrm{AD}}=\frac{C_{A}\times \Delta V^{2}}{2},$$
(4)$$P_{D}=\frac{E_{\mathrm{AD}}}{\Delta t}$$
where, *E*_AD_ is the areal energy density (Wh cm^−2^), and *P*_D_ is the areal power density (W cm^−2^).

## 3. Results

The schematic images of the fabrication method of PS/nanofiber electrode are depicted in Figure 1(ai). Copper (Cu) with a thickness of 100 nm was deposited onto cleaned PS film (2 cm × 3 cm) by sputtering (Figure 1(aii)). Afterwards, the electro-spun nanofiber was coated onto Cu-deposited PS film. According to triboelectric series, PAN and nylon are known as relatively negative and positive materials, respectively. Based on these two dielectric materials, the higher electrical output can be induced by vertical contact-separation mode [29]. The 10 wt% of PAN and nylon solutions were prepared with DMF and formic acid solvent, respectively [30,31]. The prepared polymer solutions form a nanofiber structure on the conductive PS film using electrospinning (Figure 1(aiii)). After electrospinning, a drying process is performed in an oven at 80 °C for 2 h to evaporate the solvent inside the manufactured substrate. After drying, the wrinkled PS film was fabricated by heating to 140 °C for 1 min, which is higher than the glass transition temperature of the PS. Through the glass transition of PS, the size of the wrinkled PS film was reduced to 1 cm × 1.5 cm. Herein, the as-prepared electrodes are denoted as W-PAN/PS and W-Nylon/PS, respectively (Figure 1(aiv).

The SEM images are represented with top- and cross-sections of W-PAN/PS electrode (in Figure 1b). (i) Although the PS substrate is wrinkled, the designed nanofiber structure is not destroyed by maintaining its original shape. (ii) As shown in the cross-section image, the designed nanofibers were formed onto the wrinkled PS film. Herein, the highly effective surface area of the electrode was manufactured through micro-nano hybrid structure. In detail, the micro- and nano-morphologies were created from the wrinkled PS and the electrospun nanofibers, respectively.

The schematic image of the final W-TENG structure, composed of W-PAN/PS and W-Nylon/PS, was represented in Figure 1c. According to this structure, nylon and PAN materials can vertically contact and separate to generate electrical energy. Here, an electric wire was connected to the deposited Cu to measure the electrical output produced by TENG.

The detailed working mechanism of the W-TENG is represented in Figure 1d. The electrical energy of W-TENG can be generated by the triboelectric series and electrostatic effect. Due to the triboelectric series, as two materials of nylon and PAN are contacted by external force, the nylon and PAN was positively and negatively charged, respectively (Figure 1(di)). When two materials are released by the reduced external force, a negative charge moves from the W-PAN/PS electrode to the W-Nylon/PS electrode (Figure 1(dii). When the gap between the two materials is maximized, the applied voltage between the two electrodes is maximized (Figure 1(diii). When the two materials come closer to external pressure again, negative charges induced to the electrode move in the opposite direction and current is generated in the opposite direction (Figure 1(div)). This process is iterative and produce alternating current.

Figure 2a,b demonstrate the comparisons of electrical output (*V*_OC_ and *I*_SC_) between non-wrinkled PS nanofiber TENG (NW-TENG) and wrinkled PS nanofiber TENG (W-TENG). Under the frictional condition of 1 Hz and 10 N, the 15 V and 1.125 V of *V*_OC_ was generated by W-TENG and NW-TENG, respectively (Figure 2a). Additionally, the 300 nA and 8 nA of *I*_SC_ was generated by W-TENG and NW-TENG, respectively (Figure 2b). As a result, the W-TENG produced 13 times higher *V*_OC_ and 37.5 times higher *I*_SC_ than NW-TENG.

According to the following Formulas (5) and (6), the electrical output of *V*_OC_ and *I*_SC_ can be calculated with *x*(*t*) of the time-dependent distance between the two triboelectric layers, *ε*_0_ of the permittivity of free space, *d*_0_ of the effective dielectric thickness, *σ* of the triboelectric charge density, A of the surface area of the electrode, and *v*(*t*) of the relative mechanical movement speed [32].
(5)$$V_{\mathrm{OC}}=\frac{\sigma x(t)}{\varepsilon_{0}}$$
(6)$$I_{\mathrm{SC}}=\frac{A\sigma v(t)}{d_{0}}$$

With these formulas, the wrinkled PS electrode has a wider *A*, and more triboelectric charges can be induced with the increased *σ*. Then, the higher *σ* delivers the increased *V*_OC_ and *I*_SC_ values of W-TENG. The number of transferred charges from NW-TENG and W-TENG is shown in Appendix A. The much higher charge amounts of 17 nC were successfully transferred by the W-TENG compared to 4 nC of the NW-TENG, which indicates that W-TENG can generate the electrical energy nearly four times higher than the NW-TENG can do. This is inferred that the surface area of W-TENG is also nearly four times larger than that of the NW-TENG.

Figure 2c shows the electrical performance (*V*_OC_ and *I*_SC_) of W-TENG under different operation frequency ranging from 1 to 5 Hz with an applied force of 10 N. The *I*_SC_ was increased along to the increased frequency. The *I*_SC_ was proportional to the mechanical movement, so the higher the operating frequency can generate high *I*_SC_. However, since the voltage is not related to the speed of mechanical movement, it shows almost the same values at all operating frequencies. In Figure 2d the electrical performance (*V*_OC_ and *I*_SC_) of the W-TENG was shown under different operation forces (ranging from 10 N to 50 N, 1 Hz). The results indicate that the force applied to the surface of the W-TENG is related to the output performance. It means that the charge potential of the inner contact area is increased through the increase in operation force. The *V*_OC_ and *I*_SC_ of the W-TENG under different external operating forces show a linear association. Appendix A show the electrical output at various operating forces and frequencies of W-TENG and NW-TENG. In all regions, the W-TENG produces a higher electrical output than NW-TNEG. To characterize the maximum power, various load resistors between 50 kΩ and 200 MΩ were connected to the PS-TENG at a vibration frequency of 1 Hz and a vertical force of 10 N, as shown in Figure 2e. As the load resistance (*R*) increased, the voltage was augmented while the current (*I*) exhibited a reverse change. Electrical output power (*P*) was determined using the following Formula (7):(7)$$P=I^{2}R$$

With an optimal load resistance of 20 MΩ, the power density of 6.2 μW/cm^2^ was obtained.

To use the TENG as an energy source, durability and stable output even after several uses are important. After 9000 cycles of contact–separation of 3 Hz, there was little specific reduction in *V*_OC_, as depicted in Figure 2f, maintaining 93% output compared to the operate beginning. This implying the W-TENG has good durability. As a practical application of the W-TENG, 35 LEDs can be powered under operating conditions of 50 N and 3 Hz without any other external circuit (Figure 2g).

In addition, the suggested micro-nano HSC structure could improve the electrochemical performance of SCs. The schematic images of fabrication method of PS/nanofiber electrode were depicted in Figure 3a. The PAN nanofiber was fabricated at PS film (2 cm × 2 cm) using electrospinning (Figure 3(ai,aii). After drying, the substrate was heated to 140 °C for 1 min, which is higher than the glass transition temperature of the PS, fabrication the wrinkled PS film (Figure 3(aiii). The PS film is reduced to 1 cm × 1 cm. After the substrate was shrunk, the electrospun PANI was coated onto the surface of the substrate to increase the conductivity of electrode and redox material for supercapacitor (Figure 3(aiv). The as-prepared W-PS substrate, electrospun PAN membrane, and W-PAN/PS substrate was first immersed in 45 mL of 1 M HCl solution containing 0.03 M aniline for the sufficient adsorption of aniline by PAN nanofiber or PS film. Then, another 10 mL 1 M HCl solution containing 0.015 M APS was slowly added into the above solution to initiate the polymer solution. The aniline was used for polymerization 10 to 40 min at room temperature. The as-prepared PANI-coated substrates were washed with DI water. The cleaned substrate was used after drying in a vacuum oven at 80 °C for 4 h. The PANI-coated wrinkled PAN/PS substrate is called a W-PAN/PS@PANI electrode.

Figure 3b shows the photograph of W-PAN/PS@PANI electrode. The surface of the electrode has turned dark green, confirming that the electrode surface was well coated with PANI. As shown in Figure 3c, the X-ray diffraction (XRD) characterization was conducted to examine further material analysis of synthesized electrodes, and objectively, the synthesis of PANI was confirmed. In W-PAN/PS@PANI electrode, the well-defined diffraction peaks observed at 2θ values of 28° are indexed to the (322) planes. It means the PANI was well coated surface of W-PAN/PS substrate [33].

The Figure 3d demonstrates the dependency between the PANI-coated time and surface conductivity of the electrode. As PANI coating increases, the conductivity of the electrode increases with lower resistance values [34]. The conductivity of an electrode is related to the performance of supercapacitor, as the charge of the supercapacitor is charged through ion transport and ion intercalation with the active material. So, using an electrode with high conductivity could improve the performance of supercapacitor. The SEM images of PANI-coated PAN nanofiber at different coating time is demonstrated in Appendix A. As the coating time of PANI was increased until 40 min, the amount of coated PANI was increased on the surface of the PAN nanofiber, which improved the surface conductivity of the fabricated electrode. The surface conductivity was not significantly changed after coating time longer than 40 min. Hence, the coating time of PANI was fixed at 40 min.

To study the electrochemical performance of the micro-nano hybrid electrode, a three-electrode configuration was utilized for the comparisons of the electrochemical performance of synthesized electrodes. Using 1.0 M of KOH aqueous solution as the electrolyte, the cyclic voltammetry (CV) and galvanostatic charge–discharge (GCD) plots of the PANI-coated PAN membrane (PAN@PANI), PANI-coated wrinkled PS film (W-PS@PANI), and W-PAN/PS@PANI electrode were demonstrated in Figure 3e,f. The larger integral area of CV graph and longer charge–discharge time of GCD curves indicate the higher capacity of each electrode. From these results, the W-PS/PAN@PANI electrode exhibits the higher electrochemical property than the PAN@PANI electrode. For further studies, the CV and the GCD was measured under various scan rates and different current densities, respectively. The measured CV and GCD results of the W-PS/PAN@PANI electrode is shown in Figure 3g,h, respectively.

Based on the measured GCD results, the calculated areal capacitance values are shown in Figure 4a. At 0.1 mA/cm^2^ of current density, the areal capacitance of PAN@PANI, W-PS@PANI, and W-PS/PAN@PANI electrode were 2.1, 3.3 and 4.8 F/cm^2^, respectively. The abovementioned W-PS/PAN@PANI electrode showed the highest areal capacity value because it had the largest effective area compared to other electrodes.

To investigate the charge transfer resistance and electrolyte resistance, the EIS measurement (from 10,000 to 0.01 Hz) was studied in Figure 4b. It is well known that a smaller diameter (at high frequencies) and a straight line with a higher slope (at low frequencies) may lead to more excellent chemical performance [35]. Therefore, the W-PS/PAN@PANI electrode showed more conductivity and excellent electrochemical performance than W-PS@PANI electrode. Figure 4c,d demonstrates the detailed electrochemical performance of CV curves recorded at various scan rates (from 10 to 100 mV/s) and GCD cycles studied in various current densities (from 1 to 0.1 mA/cm^2^), respectively. The inset figure of Figure 4c is a schematic diagram of hybrid supercapacitor (HSC) composed of W-PAN/PS@PANI as a positive electrode and carbon cloth of a negative electrode. Based on the increased triboelectric charge density of W-TENG and electrochemical performances of as fabricated HSC, a self-powered energy conversion and storage system was developed to confirm the properties of an energy integrated system. In Figure 4c, the quasi-rectangular shapes of CV were maintained without drastic change as well as any redox peaks. Additionally, the similar behavior of discharging curves was noticed by suggesting the great rate performance capability in Figure 4d. Finally, the cycling stability is also considered to be a crucial parameter for evaluating the performance of HSC. According to the demonstrated Figure 4e, the high capacitance retention value of 80% was achieved even after the long-term cycling test for 500 GCD cycles. As shown in Figure 4f, the Ragone plot with calculated power density and energy density illustrates the E-P relationship. From these results, a power density of 6.25 μW/cm^2^ at an energy density of 2310 μWh/cm^2^ and a power density of 62.5 μW/cm^2^ at an energy density of 315 μWh/cm^2^ were obtained, which is higher than those with previously reported [36,37,38,39].

Figure 5a shows circuit diagram of energy conversion and energy storage system. For energy generation and the establishment of an integrated energy storage system, the previously produced W-TENG and HSC using W-PAN/PS@PANI electrode were combined using a rectifier. Here, alternating current (AC) power generated by W-TENG was successfully converted into DC through a rectifier. The converted electrical power from the W-TENG was charged to the fabricated HSC, utilized to turn on a small electronic device as well as a commercial green LED. The self-charging profiles of the integrated system under various input force was depicted in Figure 5b. In Figure 5b the effect of varying magnitude of applied frequency on the charging efficiency was shown. The rate of charging was gradually in the increase of input force. As a result, the 100, 350 and 930 mV were successfully charged with the input force ranging from 1 Hz to 5 Hz at 10 N. Finally, the device can be charged to about 0.9 V within only 115 s with the periodical contact–separation under 10 N of input force and 5 Hz of frequency. Based on the excellent charging efficiency of the proposed integrated system, a demonstration of a feasible application was demonstrated by lighting commercial LEDs. The stored electrical energy generated by the PS-TENG can bright a green LED (Figure 5c). As shown in Figure 5d, a commercial thermo-hygrometer was successfully operated by the fabricated HSC, which stores the electrical energy generated from the W-TENG. Herein, the electrical energy harvested by continuous contact/separation of W-TENG was utilized as the input source to power up a commercial small electronic device. With the charged electrical power in the HSC, the display of a thermo-hygrometer was illuminated. This feasibility of the power supply from our hybrid energy systems demonstrates the broaden applicability in the field of self-powered electronic devices. Based on the above micro-nano hybrid structure characteristics, a self-powered storage energy system capable of generating electricity with high efficiency from external mechanical movement and successfully charging was manufactured. The feasibility of self-actuation of integrated devices has been successfully demonstrated as practical applications for sustainable wearable electronics and small electronic devices.

## 4. Discussion

In summary, we successfully fabricated a micro-nano hybrid system by integrating an electrospun nanofiber as well as a glass-transitioned PS film. Along to the improved surface area of as-prepared electrodes, the electrical and electrochemical performance of TENG and SC was dramatically enhanced compared to bare electrodes. In the case of TENG, Cu was deposited on the PS film for conductivity, and then the dielectric layer was fabricated by electrospinning PAN and nylon polymer. With the micro-nano hybrid structure generate increase of triboelectric charge density, the W-TENG could produce 13 times and 37.5 times higher of *V*_OC_ and *I*_SC_ than the values of NW-TENG. Finally, high power density of 6.2 μW/cm^2^ was recorded at 20 MΩ. In the case of fabricated SC, PAN nanofibers are formed on the PS film through electrospinning, and then heat is applied to fabricate a micro-nano hybrid structure. Then, for use as an electrode of a capacitor, PANI, which has conductivity and causes redox reaction, is coated. The fabricated W-PAN/PS@PANI electrode had a higher surface area and conductivity than the PAN@PANI and W-PS@PANI electrodes and showed higher electrochemical performance. HSC with carbon fiber and W-PAN/PS@PANI electrode was found to exhibit a power density of 6.25 μW/cm^2^ at an energy density of 2310 μWh/cm^2^ and a power density of 62.5 μW/cm^2^ at an energy density of 315 μWh/cm^2^. In addition, it was demonstrated that energy production and storage are possible at the same time by integrating a TENG with a micro-nano hybrid structure and a supercapacitor. Through this, the micro-nano hybrid structure presents great potential for application to energy production and storage systems.

## Figures and Tables

**Figure 1 micromachines-13-00380-f001:**
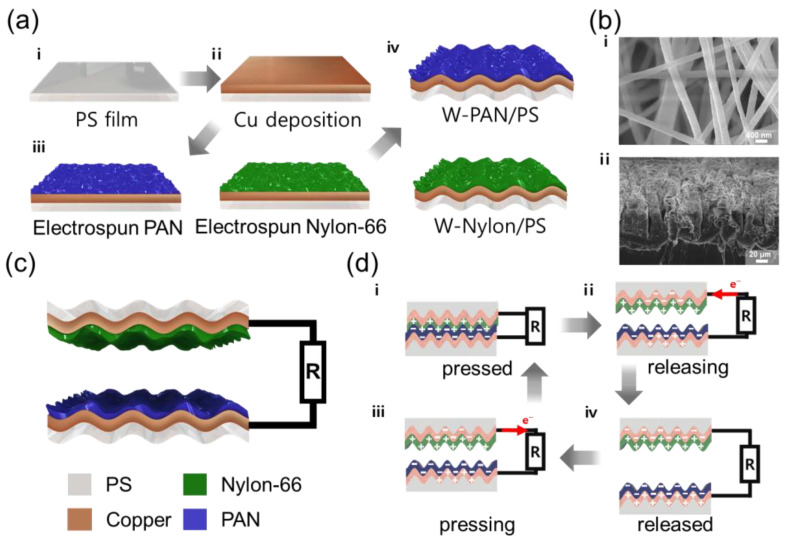
(**a**) Fabrication of W-PAN/PS and W-Nylon/PS electrode. (**b**) Field emission scanning electron microscope (FE-SEM) image of W-PAN/PS. (**c**) Schematic diagram of wrinkled TENG (W-TENG). (**d**) Schematic diagram of the electricity-generation process of W-TENG.

**Figure 2 micromachines-13-00380-f002:**
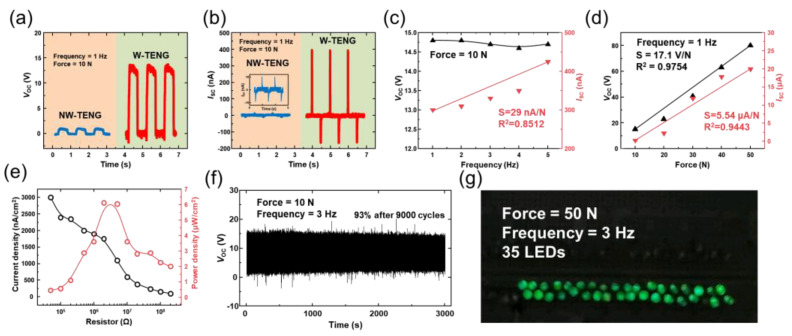
Electrical characteristics of the non-wrinkled TENG (NW-TENG) and wrinkled TENG: (**a**) *V*_OC_ and (**b**) *I*_SC_. (**c**) Frequency-dependence of W-TENG. (**d**) Force-dependence of W-TENG. (**e**) Dependence of the load resistance on current density and output power of W-TENG. (**f**) Durability of W-TENG. (**g**) Serial connection of LEDs with W-TENG.

**Figure 3 micromachines-13-00380-f003:**
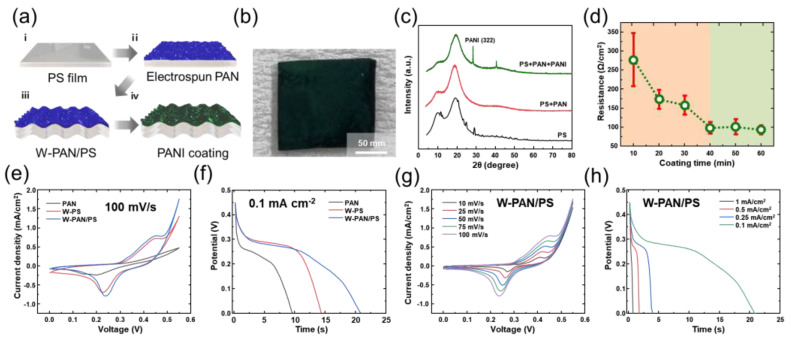
(**a**) Fabrication of wrinkled PANI-coated PS/PAN (W-PAN/PS@PANI) substrate for supercapacitor. (**b**) Photograph of W-PAN/PS@PANI electrode. (**c**) X-ray diffraction (XRD) analysis (**d**) The sheet resistance of PANI with various coating time. (**e**) CV curve measured with scan rate of 100 mV/s. (**f**) GCD curve measured with current density of 0.1 mA/cm^2^. (**g**) CV curves measured at varied scan rates from 10 to 100 mV/s. (**h**) GCD curves measured at varied current densities from 0.1 to 1 mA/cm^2^.

**Figure 4 micromachines-13-00380-f004:**
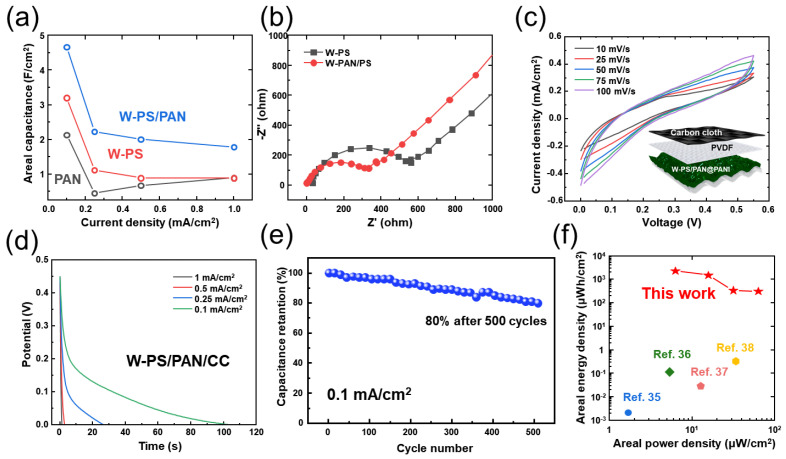
(**a**) Areal capacitance of PAN, W-PS, and W-PS/PAN. (**b**) Nyquist curves of W-PS and W-PS/PAN. (**c**) CV curves measured at varied scan rates from 10 to 100 mV/s, and (**d**) GCD curves measured at varied current densities from 0.1 to 1 mA/cm^2^. (inset) Schematic diagram of hybrid supercapacitor (HSC). (**e**) Cycling stability of W-PS/PAN at a current density of 0.1 mA/cm^2^. (**f**) The areal energy and power density of W-PS/PAN/CC supercapacitor compared with those of previously published works.

**Figure 5 micromachines-13-00380-f005:**
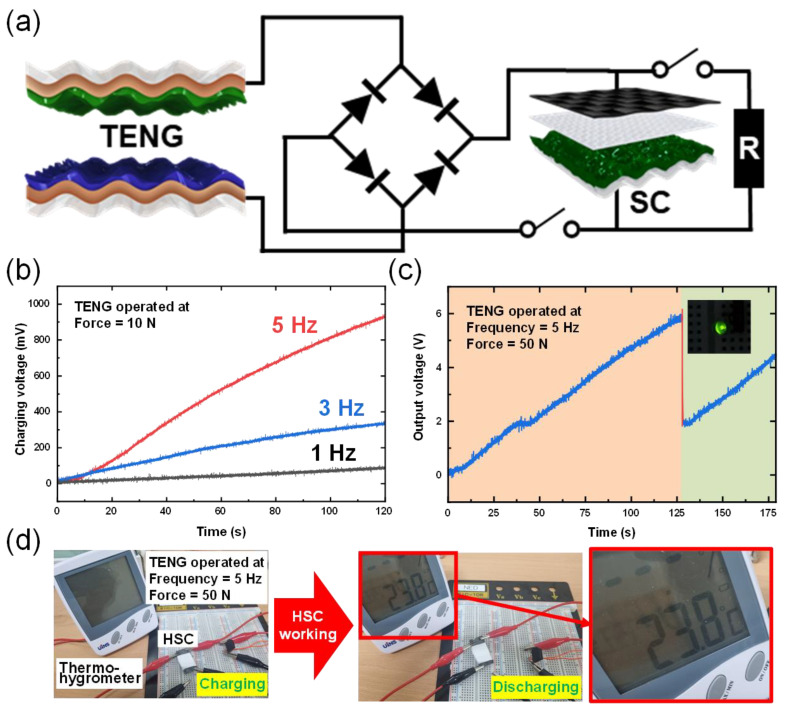
(**a**) Circuit diagram of energy generate and energy storage system. (**b**) The charging efficiency under various input frequencies (form 1 Hz to 5 Hz) of W-TENG. (**c**) The demonstration of the feasible application by lighting-up a commercial LED. (**d**) Real-time operation of the thermo-hygrometer using the hybrid energy system.

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
