# Peer review of "Electrospun Nanofiber Covered Polystyrene Micro-Nano Hybrid Structures for Triboelectric Nanogenerator and Supercapacitor"

_micromachines, 2022, doi:10.3390/mi13030380_

Round 1
Reviewer 1 Report
A micro-nano structured design of electrodes with PS, PAN, Nylon, and PANI composites was developed for achieving high performance TENG in current interesting manuscript. Comments and suggestions are as follows.
(1) Pls provide the thickness of the PS film.
(2) In Figure 2, what are the reasons for selecting 10N, 3Hz and 50N, 3Hz for (f) and (g), as 10 N, 1Hz is applied for other figures.
(3) Pls carefully check the unit of the density (m or micro) in the whole manuscript (figure and text). e.g. unit for power density and energy density in Figure 4f seems wrong.
(4) Small mistakes.
Ref. 6, the name of professor Zhonglin Wang is wrong.
Ref. 13, the name of the first author is missing.
P7, second paragraph, "after shirked (?) substrate".
Author Response
We are sincerely grateful to the editor and the reviewers for all the valuable comments and advices. Revisions have been carefully conducted with our best effort and point-by-point responses are given as follows. The sentences outside of the rectangular boxes are the comments from the reviewer and the sentences inside of the rectangular boxes are our responses to the reviewers’ comments.

Reviewer 2 Report
This paper reported the electrospun nano-fiber covered the polystyrene micro-nano hybrid structures to increase the effective surface area of electrodes for triboelectric nanogenerators and supercapacitors with the simple fabrication process. The electrical characteristic of Wrinkled TENG and Hybrid supercapacitor were investigated. The research contents are beneficial for the development of the self-powered energy storage system. After carefully reviewing this manuscript, I think it's worthy to be published in this journal after addressing the following concerns.
- Is there the optimal time to get the higher surface of W-PAN/PS. It seems that the sheet resistance keeps decreasing after 40 min as shown in figure 3d
- Did the authors measure the transferred charge of the W-TENG and compare it with the NW-TENG?
- The practical application is not found in this manuscript. It would be more interest if authors add application concepts with the proposed hybrid energy system.
Author Response

(The authors gave the same response as above.)
